# Retroperitoneal Sarcomas: An Update on the Diagnostic Pathology Approach

**DOI:** 10.3390/diagnostics10090642

**Published:** 2020-08-27

**Authors:** Joon Hyuk Choi, Jae Y. Ro

**Affiliations:** 1Department of Pathology, Yeungnam University College of Medicine, Daegu 42415, Korea; 2Department of Pathology and Genomic Medicine, Houston Methodist Hospital, Weill Medical College of Cornell University, Houston, TX 77030, USA; JaeRo@houstonmethodist.org

**Keywords:** retroperitoneal space, sarcoma, pathology, liposarcoma, leiomyosarcoma

## Abstract

Retroperitoneal sarcomas are a heterogenous group of rare tumors arising in the retroperitoneum. Retroperitoneal sarcomas comprise approximately 10% of all soft tissue sarcomas. Though any soft tissue sarcoma histologic types may arise in the retroperitoneal space, liposarcoma (especially well-differentiated and dedifferentiated types) and leiomyosarcoma do so most commonly. Retroperitoneal sarcomas are diagnostically challenging, owing to their diversity and morphological overlap with other tumors arising in the retroperitoneum. An accurate diagnosis is necessary for correct management and prognostication. Herein, we provide an update on the diagnostic approach to retroperitoneal sarcomas and review their key histologic findings and differential diagnoses.

## 1. Introduction

Soft tissue sarcomas are rare malignant mesenchymal tumors that account for less than 1% of all malignant tumors. The etiology of most soft tissue tumors is unknown. In rare cases (<10%), genetic factors, environmental factors, irradiation, viral infections, and immunodeficiency may be associated with the development of soft tissue sarcomas [1]. Soft tissue tumors have been classified predominantly based on the line of differentiation—that is, which normal cell type the neoplastic cells most closely resemble [2]. The 2020 World Health Organization (WHO) classifies soft tissue tumors into 12 subtypes according to their lineage: (1) adipocytic tumors, (2) fibroblastic and myofibroblastic tumors, (3) so-called fibrohistiocytic tumors, (4) vascular tumors, (5) pericytic (perivascular) tumors, (6) smooth muscle tumors, (7) skeletal muscle tumors, (8) gastrointestinal stromal tumors, (9) chondro-osseous tumors, (10) peripheral nerve sheath tumors, (11) tumors of uncertain differentiation, and (12) undifferentiated small round cell sarcomas [3].

Primary retroperitoneal soft tissue tumors constitute a heterogenous group of neoplasms. Though benign lesions typically predominate over malignant lesions elsewhere in the body, malignant lesions of the retroperitoneum are roughly four times more frequent than benign lesions [4]. Around 10% of all sarcomas occur in the retroperitoneum. The diagnosis of retroperitoneal tumors is complicated by (1) a large number of tumor types, (2) morphological overlap between various tumor types, and (3) the increasing use of minimally invasive biopsy techniques with very limited tissue. An accurate diagnosis is crucial for correct management and prognostication. Herein, we review the diagnostic pathology approach to retroperitoneal sarcomas and their updated histological and molecular features introduced by the 2020 WHO classification of soft tissue tumors.

## 2. Diagnostic Pathology Approach to Retroperitoneal Tumors

### 2.1. Anatomy of the Retroperitoneum

The retroperitoneum is a complex anatomic compartment. The retroperitoneal space is bound superiorly by the 12th rib and vertebra, inferiorly by the sacrum and iliac crest, anteriorly by the peritoneum, posteriorly by the posterior abdominal wall, and laterally by the peripheral margin of the quadratus lumborum muscles [5]. The retroperitoneal space contains the esophagus, pancreas (except tail), duodenum (second and third parts), ascending and descending colon, rectum, adrenal glands, kidneys, ureter, aorta, inferior vena cava, lymph nodes, and nerve roots. Loose connective tissue is present between the organs. The retroperitoneal space is potentially large and, therefore, retroperitoneal tumors can grow considerably before manifesting clinical signs and symptoms.

Some sarcomas arise more commonly within the retroperitoneum than others. Sarcomas that occur relatively commonly versus rarely are shown in Table 1. The most common retroperitoneal sarcomas are liposarcoma (specifically well-differentiated and dedifferentiated subtypes) and leiomyosarcoma [6]. Sarcomas arising within the retroperitoneum may be infiltrative and therefore difficult to manage surgically. A thorough understanding of the anatomy of the retroperitoneum is necessary to establish a working differential diagnosis and aid preoperative planning.

### 2.2. Clinical and Imaging Considerations

The majority of retroperitoneal sarcomas are large at presentation; indeed, nearly 50% are larger than 20 cm at diagnosis [7]. Symptoms secondary to retroperitoneal lesions appear late in the course of disease and are associated with the displacement of organs and obstructive phenomena [8]. Since retroperitoneal sarcomas often involve vital structures, complete surgical resection is often not possible. Therefore, the overall prognosis of retroperitoneal sarcomas is worse than that of sarcomas in the extremities. A variety of imaging techniques, including ultrasonography, computed tomography (CT), and magnetic resonance imaging (MRI), may be used to assess retroperitoneal tumors. Radiologic imaging plays a key role in the evaluation of tumors arising in this region. Imaging provides useful information for identifying, localizing, and characterizing the tumors, formulating the differential diagnosis, and planning for surgical resection.

Pathologists should be aware of the patient’s clinical history that might help point towards a certain differential diagnosis. Radiation-associated sarcomas represent approximately 5% of all sarcomas. The most common histologic subtypes of radiation-associated sarcomas include angiosarcoma, leiomyosarcoma, extraskeletal osteosarcoma, malignant peripheral nerve sheath tumor (MPNST), and undifferentiated sarcoma [9]. Some types of soft tissue tumors occur on a familial or inherited basis. For example, in around 5–10% of patients with neurofibromatosis type 1 (NF1), MPNSTs develop, usually in a benign nerve sheath tumor [10]. A combination of the clinical history and radiographic findings can be helpful in the differential diagnosis of retroperitoneal tumors [11,12].

### 2.3. Retroperitoneal Tumor Specimen Handling

Orienting resected specimens of retroperitoneal tumors is often complicated and proper handling of resection specimens by pathologists is crucial for diagnostic accuracy. When a specimen is submitted, it is necessary to discuss the case with the surgeon to orient the specimen correctly and to identify the true margins [13]. Closest resection margins should be inked. The margin sections are taken perpendicular to the inked surface to assess the distance from the tumor to margin. An appropriate number of blocks from the tumor is required and generally determined by a useful rule of thumb that suggests one section be submitted for every 1 cm of the maximum diameter of the tumor [14]. In addition, it is helpful to reserve fresh tumor tissue for electron microscopy, cytogenetic analysis, and other special studies.

Areas with differences in the gross appearance (e.g., hemorrhagic, necrotic, fleshy, fibrous, mucoid, or gritty) are of particular importance to pathologists. Furthermore, sections from any foci within the tumor that look different from other areas of the tumor should be submitted. When approaching a retroperitoneal pleomorphic sarcoma, extensive sampling should be performed to search for diagnostic clues (e.g., lipoblasts, myxoid stroma, or osteoid matrix) [15].

### 2.4. Histologic Evaluation

The first and most important step in reaching a correct diagnosis is careful scrutiny of conventionally stained sections at low-power magnification [16]. A pattern-based approach is a useful technique that substantially aids the diagnostic process. At low-power magnification, the degree of cellularity, growth (architectural) pattern, tumor cell appearances, and stroma characteristics should be examined. Retroperitoneal mesenchymal tumors can be categorized based on tumor cell morphology into four groups: spindle cell, epithelioid cell, round cell, and pleomorphic cell (Table 2). Growth patterns vary and include fascicular, storiform, palisading, rosettes, lobular, nests, sheets, and biphasic. Tumor cell morphology and growth patterns are helpful for narrowing potential differential diagnoses.

Histological assessment of retroperitoneal mesenchymal tumors remains a challenge as their morphology frequently overlaps with several other tumors and because some lack distinguishing immunohistochemical markers. Malignant soft tissue tumors are generally characterized by nuclear atypia, pleomorphism, increased mitoses, granular tumor necrosis [17], and infiltrative margins. Some tumors have characteristic nuclear (e.g., cigar-shaped, blunt-ended nuclei in leiomyosarcoma), cytoplasmic (e.g., clear or eosinophilic granular in PEComa), and stromal features (e.g., prominent inflammatory cell infiltration in inflammatory well-differentiated liposarcoma). The presence of lipoblasts and osteoid matrix can be a diagnostic clue in liposarcoma and osteosarcoma, respectively. An understanding of tumor cell morphology, growth patterns, cytoplasmic features, and stromal features can facilitate proper diagnosis. Histological typing alone does not provide sufficient information for predicting the clinical course of disease [1]. Grading and staging are the two most important prognostic and predictive factors. The FNCLCC (Fédération nationale des centres de lutte contre le cancer) grading and TNM (tumor, lymph node, metastasis) staging system for sarcomas are widely used [6,18].

### 2.5. Immunohistochemistry

Many types of soft tissue tumors lack distinctive morphological features and have an uncertain line of differentiation. Immunohistochemistry plays a critical role in identifying the line of differentiation and serves as a surrogate for underlying molecular genetic alterations.

Although many immunohistochemical markers have limited specificities, antibodies directed against protein correlates of specific molecular genetic alterations have been recently developed [19,20].

Useful immunohistochemical markers for retroperitoneal sarcomas include SMA, desmin, myogenin, CD34, S100 protein, MDM2, CDK4, STAT6, ALK, CD99, H3K27me3, NKX2.2, TLE1, SOX10, melanocytic markers (e.g., HMB-45, melan-A), cyclin D1, and epithelial markers (e.g., cytokeratin, EMA) (Table 3). An appropriate immunohistochemical panel is necessary for accurate diagnosis. Pathologists should interpret immunohistochemical findings carefully in the context of clinical and histological findings.

### 2.6. Molecular Testing

Soft tissue sarcomas can be divided broadly into two genetic classes: (1) simple karyotype sarcomas associated with a recurrent mutation or translocation (e.g., synovial sarcoma, Ewing sarcoma) and (2) complex karyotype sarcomas with numerous chromosomal aberrations but a general lack of recurrent mutations (e.g., pleomorphic liposarcoma, leiomyosarcoma, MPNST) [21,22,23]. Specific genetic alterations identified in retroperitoneal sarcomas are summarized in Table 4 [24,25,26,27,28,29,30].

In recent years, there has been marked progression in the application of molecular testing to the diagnosis of soft tissue tumors. New development tools such as comparative genomic hybridization, gene expression arrays, and next-generation sequencing make important contributions not only to our biological understanding but also to classification, prognostication, and treatment approaches for soft tissue sarcomas [31]. Interpretation of molecular testing can be sometimes difficult. Paramount is the fact that molecular testing cannot be used in isolation. The selection of any particular molecular testing should be on the basis of a specific differential diagnosis and relevant pretest probabilities [31]. In the setting of unusual histologic and immunohistochemical findings, molecular testing plays a critical role in the differential diagnosis.

## 3. Relatively Common Retroperitoneal Sarcomas

### 3.1. Liposarcoma

Liposarcoma is a soft tissue sarcoma with lipogenic differentiation and varying biological behavior, ranging from locally aggressive to metastasizing. According to the 2020 WHO classification of soft tissue and bone tumors, liposarcomas are classified into five major subtypes: (1) atypical lipomatous tumor/well-differentiated, (2) dedifferentiated, (3) myxoid, (4) pleomorphic, and (5) myxoid pleomorphic [32]. All liposarcoma subtypes develop in the retroperitoneum. The majority of retroperitoneal liposarcomas are well-differentiated liposarcomas and dedifferentiated subtypes. Pleomorphic and myxoid liposarcomas are rare in the retroperitoneum.

#### 3.1.1. Well-Differentiated Liposarcoma

Atypical lipomatous tumor (ALT)/well-differentiated liposarcoma (WDLPS) is a locally aggressive but not metastasizing mesenchymal neoplasm composed either entirely or partly of adipocytic proliferation, showing at least focal nuclear atypia in both adipocytes and stromal cells [33]. ALT/WDLPS accounts for approximately 40–45% of all liposarcomas. It most frequently occurs in the deep soft tissue of the proximal extremities and trunk. The retroperitoneum is also commonly involved. ALT and WDLPS are synonyms describing lesions that are morphologically and genetically identical. Use of either term is determined principally by tumor location and resectability [33]. Tumors arising in the retroperitoneum are very difficult to resect completely and best classified as WDLPS. Clinically, retroperitoneal lesions are often asymptomatic until the tumor has exceeded 20 cm in diameter.

Histologically, ALT/WDLPS is divided into three subtypes: (1) adipocytic (lipoma-like), (2) sclerosing, and (3) inflammatory [34]. The presence of more than one morphological pattern in the same lesion is common, particularly in retroperitoneal tumors. Adipocytic (lipoma-like) WDLPS shows mature adipocytes with variation in size along with nuclear hyperchromasia and atypia of the adipocytes and stromal cells. Lipoblasts can also be found. Sclerosing WDLPS shows scattered bizarre stromal cells with nuclear hyperchromasia in an extensively collagenous stroma. Inflammatory WDLPS occurs most often in the retroperitoneum and has dense, chronic inflammatory cell infiltrate with scattered, atypical, often bizarre multinucleated stromal cells (Figure 1) [35]. Immunohistochemically, nuclear expression of MDM2 and CDK4 is present in most cases. FISH evaluation for amplification of *MDM2* can help confirm the diagnosis [36].

Differential diagnoses of WDLPS include retroperitoneal lipoma, atypical spindle cell/pleomorphic lipomatous tumor, lipomatous (fat-forming) solitary fibrous tumor (SFT), and dedifferentiated liposarcoma. Benign lipomas of the retroperitoneum are extremely rare and are circumscribed, multilobulated, and composed of normal-appearing mature adipose tissue, with no cytologic atypia [37]. Diagnosis of retroperitoneal lipoma should be made after very careful histologic, cytogenetic, and molecular analyses. Atypical spindle cell/pleomorphic lipomatous tumors rarely occur in the retroperitoneum and are negative for MDM2 and CDK4 [38]. Lipomatous (fat-forming) SFTs are positive for STAT6 and negative for MDM2 and CDK4 [39]. Dedifferentiated liposarcomas show cellular, usually non-lipogenic sarcomas with a wide morphological spectrum. Inflammatory WDLPS should be distinguished from inflammatory myofibroblastic tumor, Castleman disease, and hematologic malignancy. The presence of MDM2-positive, atypical stromal cells is a useful diagnostic clue.

#### 3.1.2. Dedifferentiated Liposarcoma

Dedifferentiated liposarcoma (DDLPS) is an ALT/WDLPS showing progression, either in the primary or in a recurrence, to a (usually non-lipogenic) sarcoma of variable histological grade [40]. A well-differentiated component may not be found. Rarely, the high-grade component may be lipogenic. Approximately 90% of cases arise de novo, and 10% develop in recurrences. The most common site of DDLPS is the retroperitoneum. Other locations include the spermatic cord, mediastinum, head, neck, and trunk [41,42]. DDLPS shows significant genetic overlap with ALT/WDLPS, with amplification of *MDM2* and *CDK4* [43]. Some genomic features appear to be more often related to DDLPS, although not restricted to DDLPS, such as the amplification of *JUN* (1p32.1), *TERT* (5p15.33), *CPM*, *MAP3K5*, and other genes from the 6q21–q24 region [40]. Clinically, retroperitoneal DDLPSs are frequently found incidentally as large painless masses.

Histologically, DDLPS demonstrates an abrupt or gradual transition from WDLPS to non-lipogenic sarcoma, which, in most cases, is high-grade (Figure 2). Dedifferentiated areas vary histologically but frequently resemble undifferentiated pleomorphic sarcoma or intermediate- to high-grade myxofibrosarcoma [41,44]. Cases with low-grade dedifferentiation are increasing [42]. DDLPSs exhibit heterologous differentiation in around 5–10% of cases [45]. Most often, the line of heterologous differentiation is rhabdomyosarcomatous, leiomyosarcomatous, or osteosarcomatous. A distinctive neural-like pattern or meningothelial-like whorling pattern is present in some cases [46]. The dedifferentiated component of DDLPS can show homologous lipoblastic differentiation (Figure 3) [47]. WDLPS and DDLPS can show myogenic differentiation [48]. Immunohistochemically, the tumor cells demonstrate nuclear expression of MDM2 and CDK4.

Differential diagnoses of DDLPS are broad as there is morphological heterogeneity and they include WDLPS, pleomorphic liposarcoma, pleomorphic leiomyosarcoma, MPNST, and undifferentiated pleomorphic sarcoma (UPS). WDLPSs lack an overtly non-lipogenic sarcoma area. DDLPSs with homologous lipoblastic differentiation can closely resemble pleomorphic liposarcoma. Pleomorphic liposarcomas lack components of WDLPS and show no *MDM2* amplification. Pleomorphic leiomyosarcomas lack components of WDLPS and are positive for SMA and desmin but negative for MDM2 and CDK4. MPNSTs show fascicles of spindle cells with wavy, tapering nuclei. MDM2 expression can be seen in MPNST [49]. High-level *MDM2* amplification strongly suggests DDLPS over MPNST. UPSs lack components of WDLPS and show no *MDM2* amplification. Most tumors resembling UPSs in the retroperitoneum are DDLPSs [50]. Therefore, extensive sampling with immunohistochemistry for MDM2 and CDK4, and FISH for *MDM2* amplification, are helpful to make a proper diagnosis of DDLPS with differential diagnosis.

#### 3.1.3. Pleomorphic Liposarcoma

Pleomorphic liposarcoma (PLPS) is a pleomorphic, high-grade sarcoma containing a variable number of pleomorphic lipoblasts [51]. No areas of ALT/WDLPS or other lines of differentiation are present. PLPS is a rare subtype of LPS, accounting for ~5% of all LPSs. Most cases occur in old adults, with peak incidence in the seventh decade of life. Men are affected slightly more often than women. PLPS most often arises in the lower and upper extremities [52,53,54]. The retroperitoneum and trunk are less commonly affected. PLPSs share very similar genomic profiles with other high-grade pleomorphic sarcomas and exhibit complex molecular profiles with numerous chromosomal imbalances [55,56]. Frequently mutated genes include *TP53* (17% of cases) and *NF1* (8% of cases) [57].

Histologically, PLPS shows a varying proportion of pleomorphic lipoblasts in a background of high-grade, usually pleomorphic undifferentiated sarcoma features. The presence of lipoblasts is necessary for accurate diagnosis of PLPS. Lipoblasts have irregular, hyperchromatic, scalloped nuclei with univacuolated or multivacuolated cytoplasm. Lipoblasts can be few in number and therefore adequate sampling is required to identify them. An intermediate to high-grade myxofibrosarcoma-like component is present in some cases [58]. An epithelioid morphology is seen in around 25% of cases [59]. Immunohistochemically, S100 protein is positive in adipocytes and may be useful for highlighting lipoblasts in tumors mimicking UPS. Staining for MDM2 and CDK4 is typically negative. Epithelioid variants of PLPS are frequently positive for cytokeratin and melan-A [58].

Differential diagnoses of PLPS include WDLPS, DDLPS, myxofibrosarcoma, UPS, and carcinoma. WDLPSs lack pleomorphic lipoblasts and non-lipogenic pleomorphic sarcoma components and are positive for MDM2 and CDK4. DDLPSs frequently contain components of WDLPS and are positive for MDM2 and CDK4 and show *MDM2* amplification. Myxofibrosarcomas are extremely rare in the retroperitoneum and lack lipoblastic differentiation. UPSs show no evidence of lipoblastic differentiation and are negative for MDM2 and CDK4. Epithelioid variants of PLPS may be mistaken for poorly differentiated carcinomas, such as adrenal cortical and renal cell carcinoma. Carcinomas are positive for epithelial markers (e.g., cytokeratin, EMA). Recognition of pleomorphic lipoblasts is the most important diagnostic clue for PLPS. Lipoblast-like cells may be seen in a variety of conditions, and failure to recognize an appropriate histologic background can lead to an erroneous diagnosis of LPS [60].

#### 3.1.4. Myxoid Liposarcoma

Myxoid liposarcoma (MLPS) is a malignant tumor composed of uniform, round to ovoid cells and a variable number of small lipoblasts [61]. MLPS is set in a myxoid stroma with a branching capillary vasculature and accounts for approximately 20–30% of LPSs. The peak incidence is in the fourth and fifth decades of life. MLPSs typically develop within the deep soft tissues of the extremities. Distant metastases develop in approximately 30–60% of cases. Primary retroperitoneal MLPSs are extremely rare [62,63]. Most cases of MLPS in the retroperitoneum represent a metastasis. MLPSs show a specific t(12;16)(q13;p11) with *FUS-DDIT3* fusion gene in most cases (90–95%) and rarely t(12;22)(q13;q12) with *EWSR1-DDIT3* fusion gene [64]. Clinically, MLPSs present as large, painless masses.

Histologically, MLPS is a moderately cellular, lobulated tumor composed of uniform, small, ovoid cells in a myxoid stroma with variable numbers of small lipoblasts [61]. Increased cellularity is typically present in the periphery of the lobules. A characteristic plexiform, delicately arborizing, capillary network (chicken wire pattern) is present. The transitional area with modest increase in cellularity should not be interpreted as a round cell change. High-grade MLPS (>5% round cell component) shows cellular overlapping, larger and more hyperchromatic nuclei, and increased mitotic activity. High-grade tumors have a higher risk of metastasis or death from the disease [65,66]. Immunohistochemically, S100 protein may highlight lipoblasts or show focal expression in round cell areas.

Differential diagnoses of MLPSs include WDLPS, myxofibrosarcoma, poorly differentiated synovial sarcoma, and extraskeletal Ewing sarcoma (ES). WDLPSs show scattered enlarged hyperchromatic stromal cells and are positive for MDM2 and CDK4. In adipocytic tumors with myxoid stroma arising in the retroperitoneum, the possibility of WDLPS should be first considered. Myxofibrosarcomas show nuclear atypia and pleomorphism and lack true lipoblasts, with a curvilinear vascular pattern rather than chicken-wire vascular pattern. High-grade MLPS requires a differential diagnosis from poorly differentiated synovial sarcomas that lack lipoblasts and an arborizing capillary vascular pattern and are diffusely positive for TLE1 and SS18-SSX [67]. Extraskeletal ESs lack lipoblasts and an arborizing capillary vascular pattern and show strong membranous positivity for CD99 and nuclear positivity for NKX2.2.

### 3.2. Leiomyosarcoma

Leiomyosarcoma is a malignant neoplasm composed of cells showing smooth muscle differentiation [68]. Soft tissue leiomyosarcomas commonly arise in the extremities (particularly the lower extremities), retroperitoneum, abdomen, pelvis, and trunk [69,70]. Leiomyosarcoma is the second most common sarcoma in the retroperitoneum. Retroperitoneal leiomyosarcoma usually occurs in middle-aged or older adults, with a female predominance. A subset of retroperitoneal leiomyosarcomas arises from large blood vessels, including the inferior vena cava and renal vein [71]. Retroperitoneal leiomyosarcomas are typically large and often difficult to excise with clear margins.

Histologically, leiomyosarcoma typically shows fascicles of spindle-shaped tumor cells with blunt-ended nuclei and moderate to abundant, brightly eosinophilic fibrillary cytoplasm (Figure 4). Higher-grade tumors exhibit nuclear atypia, frequent mitotic figures, and granular tumor necrosis. Histologic variants include inflammatory leiomyosarcoma [72], Epstein–Barr virus-associated smooth muscle tumors [73], myxoid leiomyosarcoma [74], epithelioid leiomyosarcoma [75], pleomorphic leiomyosarcoma [76], and dedifferentiated leiomyosarcoma [77]. Criteria for malignancy in retroperitoneal smooth muscle tumors are still unclear [78]. Mitotic activity, nuclear atypia, and coagulative necrosis are important prognostic factors. Retroperitoneal smooth muscle tumors showing nuclear atypia with any mitosis can be considered as malignancy [79]. Retroperitoneal uterine-type smooth muscle tumors with 5 to 10 mitoses per 50 high-power fields (HPFs) and no other worrisome features can be regarded as having uncertain malignant potential [80]. ER and PR are frequently positive. Immunohistochemically, at least one myogenic marker (i.e., SMA, desmin, or caldesmon) is positive, with >70% of cases showing positivity for more than one of these markers. Tumor cells are also positive for cytokeratin and EMA in approximately 40% of cases [81].

Differential diagnoses of leiomyosarcoma include schwannoma, inflammatory myofibroblastic tumor (IMT), gastrointestinal stromal tumor (GIST), MPNST, synovial sarcoma, perivascular epithelioid cell tumor (PEComa), and UPS. Schwannomas are strongly and diffusely positive for S100 protein. IMTs show admixed chronic inflammatory cell component and are positive for ALK. GISTs usually arise from the gastrointestinal tract wall, mesentery, or omentum, are occasionally positive for SMA but are diffusely positive for CD34, CD117, and DOG1. MPNSTs are focally positive for S100 protein in <50% of cases and show loss of H3K27me3 expression. Synovial sarcomas demonstrate monomorphic tumor cells and are strongly and diffusely positive for TLE1 and SS18-SSX. PEComas show sheets of epithelioid and spindle cells and are positive for melanocytic markers (e.g., HMB-45, melan-A). UPSs lack areas of conventional leiomyosarcoma and are negative for desmin and caldesmon.

## 4. Rare Retroperitoneal Sarcomas

### 4.1. Solitary Fibrous Tumor

Solitary fibrous tumor (SFT) is a fibroblastic tumor characterized by a prominent, branching, thin-walled, dilated (staghorn) vasculature and *NAB2-STAT6* gene rearrangement [82]. SFTs may occur at any anatomical site. Extrapleural lesions are more common than pleural lesions. Around 30–40% of extrapleural SFTs arise in deep soft tissues, including the abdominal cavity, pelvis, and retroperitoneum [83,84]. SFTs most commonly affect adults, with a peak incidence between 40 and 70 years. The genetic hallmark of SFT is a paracentric inversion involving chromosome 12q, resulting in *NAB2-STAT6* gene fusion [85]. Clinically, most tumors present as slow-growing, painless masses.

Histologically, SFTs are composed of spindle to ovoid cells with indistinct, pale, eosinophilic cytoplasm that are haphazardly arranged in a collagenous stroma. Branching, staghorn-shaped (hemangiopericytomatous) blood vessels are present. There is a wide histological spectrum. Histologic variants include myxoid SFT [86], lipomatous (fat-forming) SFT [87], giant cell-rich SFT [88], and dedifferentiated SFT (Figure 5) [89]. Malignant SFTs show a high mitotic count (>4 mitoses per 10 HPFs), increased cellularity, cytological atypia, necrosis, and/or infiltrative growth. Of these features, mitoses are regarded as the most important prognostic factor. A newly described risk stratification model based on patient age, tumor size, mitotic count, and tumor necrosis more clearly delineates prognosis [90].

Rhabdomyoblastic differentiation is also described as a rare phenomenon in malignant SFT [91]. Immunohistochemically, the tumor cells exhibit strongly and diffusely positive cytoplasmic expression of CD34 and nuclear STAT6 expression; however, the expression of CD34 and STAT6 is frequently lost in dedifferentiated SFT. STAT6 is a highly sensitive and specific immunohistochemical marker for SFT [92].

Differential diagnoses of SFT include schwannoma, GIST, MPNST, synovial sarcoma, and DDLPS. Schwannomas are strongly and diffusely positive for S100 protein. GISTs are positive for CD117 and DOG1 and negative for STAT6. MPNSTs show wavy, tapering, fascicularly arranged nuclei and are focally positive for S100 protein and SOX10 in <50% and <70% of cases, respectively. Synovial sarcomas demonstrate strong and diffuse nuclear expression of TLE1 and SS18-SSX and are negative for CD34 and STAT6. Approximately 10% of DDLPSs cases express STAT6, which may be a potential pitfall in the differential diagnosis of SFTs, particularly malignant SFTs [93,94,95]. DDLPSs are positive for MDM2 and CDK4 and show *MDM2* amplification.

### 4.2. Inflammatory Myofibroblastic Tumor

Inflammatory myofibroblastic tumor (IMT) is a distinctive, rarely metastasizing neoplasm composed of myofibroblastic and fibroblastic spindle cells accompanied by inflammatory infiltrate of plasma cells, lymphocytes, and/or eosinophils [96]. IMT occurs mainly in children and young adults but can arise in older adults, with a slight female predominance. IMT shows a wide anatomic distribution. It commonly arises in the abdomen (mesentery, omentum) and retroperitoneum [97,98]. In approximately 50–60% of IMTs, the tumors harbor rearrangement of the *ALK* gene at chromosome 2p23. In *ALK*-negative IMTs, *ROS1* gene rearrangement and *ETV-NTRK3* gene fusion have been described [99,100]. Approximately 25% of extrapulmonary IMTs recur. Distant metastases are rare (<5%). ALK-negative IMTs exhibit a higher risk of metastasis than ALK-positive IMTs [101]. ALK-negative IMTs also occur in older patients and have greater nuclear pleomorphism, atypia, and atypical mitoses. Clinically, the site of origin determines symptoms.

Histologically, IMT is composed of spindle-shaped fibroblasts and myofibroblasts in a myxoid and collagenous stroma with prominent inflammatory cell infiltrate, including lymphocytes and plasma cells. The three basic histological patterns are (1) myxoid pattern characterized by loosely arranged spindle cells in an edematous myxoid stroma with abundant blood vessels, (2) hypercellular pattern showing compact spindle cell proliferation, and (3) hypocellular fibrous pattern characterized by hyalinized collagenous stroma and relatively sparse inflammatory infiltrate [96,97]. A variety of histologic patterns may be present in the same tumor. Epithelioid inflammatory myofibroblastic sarcoma (EIMS) is a distinctive IMT subtype with plump, round epithelioid or histiocytoid tumor cells (Figure 6) [102]. EIMS is associated with *RANBP2-ALK* or *RRBP1-ALK* gene rearrangement and has an aggressive clinical course. Immunohistochemically, IMTs show variable positivity for myofibroblastic markers (e.g., SMA, muscle-specific actin, and desmin). Immunoreactivity for ALK is present in 50–60% of cases. The ALK immunostaining pattern varies depending on the *ALK* fusion partner. *RANBP2-ALK* is associated with a nuclear membranous pattern.

Differential diagnoses of IMT vary according to histologic patterns and include retroperitoneal fibrosis, desmoid fibromatosis, GIST, inflammatory WDLPS, DDLPS, and leiomyosarcoma. Retroperitoneal fibrosis is a rare lesion involving soft tissues or organs in the retroperitoneum and can be classified into IgG4-related and non-IgG4-related cases [103]. It shows dense fibrosis with chronic inflammatory cell infiltrate and is negative for ALK. Desmoid fibromatosis exhibits long fascicles of bland spindle cells and nuclear β-catenin expression. GISTs are diffusely positive for CD117 and DOG1. Inflammatory WDLPSs are positive for MDM2 and CDK4. DDLPSs contain components of WDLPS and are positive for MDM2 and CDK4. Leiomyosarcomas may have a prominent inflammatory cell infiltrate but show conventional leiomyosarcoma morphology and are diffusely positive for SMA and negative for ALK.

### 4.3. Rhabdomyosarcoma

Rhabdomyosarcoma (RMS) is a malignant soft tissue tumor showing various stages in the embryonic differentiation of skeletal muscle. RMS is the most common soft tissue sarcoma in children and adolescents, with 4.5 cases per million individuals aged 0–20 years. However, it can affect patients of all ages [104]. The 2020 WHO classification of RMS is divided into embryonal, alveolar, pleomorphic, and spindle cell/sclerosing subtypes. RMS is the most common retroperitoneal sarcoma in children. The most common subtype of primary retroperitoneal RMS is embryonal RMS. Alveolar, pleomorphic, and spindle cell/sclerosing RMS subtypes rarely arise in the retroperitoneum [105,106,107]. Clinically, RMS presents with a variety of clinical symptoms, generally related to the mass effect.

Histologically, embryonal RMS is composed of spindle cells and more primitive rounded cells, variably differentiated round, strap-shaped, or tadpole-shaped eosinophilic rhabdomyoblasts with alternating areas of loose and dense cellularity (Figure 7). Histologic variants of embryonal RMS include the botryoid and anaplastic types [108]. Occasionally, embryonal RMSs show a dense cellular pattern of primitive round cells, simulating alveolar RMSs [109]. Alveolar RMS shows uniform primitive round cell morphology with an alveolar growth pattern [110]. Pleomorphic RMS is composed of sheets of large, atypical cells and frequently multinucleated polygonal, spindle-shaped, or rhabdoid cells [111]. Spindle cell/sclerosing RMS is characterized by cellular spindle cell fascicles or round tumor cells in a sclerotic collagenous stroma [112]. Immunohistochemically, the tumor cells are positive for desmin, myogenin, and MYOD1. Nuclear expression of myogenin is stronger and more uniform in alveolar RMS compared to embryonal RMS. RMSs show diffuse and strong cytoplasmic expression of WT1 [113].

Differential diagnoses of embryonal RMS include infantile fibrosarcoma, alveolar RMS, extraskeletal ES, desmoplastic small round cell tumor (DSRCT), and extrarenal rhabdoid tumor. Infantile fibrosarcomas exhibit intersecting fascicles of primitive ovoid and spindle cells, lack expression of myogenic markers, and show *ETV6-NTRK3* gene fusion [114]. Alveolar RMSs show more uniformly rounded, undifferentiated cells with larger nuclei than embryonal RMSs, with an alveolar pattern and diffuse myogenin expression. Extraskeletal ESs show diffuse membrane expression of CD99 and are negative for desmin and myogenin. DSRCTs demonstrate a desmoplastic fibrous stroma and are positive for cytokeratin, EMA, and desmin but negative for myogenin. Extrarenal rhabdoid tumors show polygonal rhabdoid cells with abundant eosinophilic cytoplasm and are positive for cytokeratin and EMA, with loss of SMARCB1 (INI1) expression.

### 4.4. Malignant Peripheral Nerve Sheath Tumor

Malignant peripheral nerve sheath tumor (MPNST) is a malignant spindle cell tumor that often arises from a peripheral nerve or a pre-existing benign nerve sheath tumor or in a patient with NF1 [115]. MPNST most commonly occurs in patients aged 20–50 years, with a broad age range. Patients with NF1 usually present at a slightly earlier age. MPNST usually arises in the trunk, extremities, head, and neck [116]. Primary retroperitoneal MPNSTs are rare. Patients with NF1 usually develop MPNSTs after a relatively long latency (10–20 years). Approximately 10% of cases are associated with prior radiation therapy [117]. Rare cases of MPNSTs may arise from schwannoma and ganglioneuroma [118,119]. Clinically, MPNST presents as an enlarging painless or painful mass.

Histologically, MPNST is typically composed of spindle cells with wavy, buckled, tapering nuclei showing a fascicular growth pattern, with alternating hypercellular and hypocellular myxoid areas (Figure 8). A branching hemangiopericytoma-like vascular pattern is often present. Perivascular accentuation of tumor cells may be a diagnostic clue. MPNST has a diverse microscopic appearance. Heterologous differentiation, including rhabdomyoblastic (malignant triton tumor), osteosarcomatous, chondrosarcomatous, and angiosarcomatous, is seen in 10–15% of MPNSTs [115]. Lipoblastic differentiation is also described as a rare phenomenon in MPNST [120]. Very rare cases may show glandular differentiation (glandular MPNST). Epithelioid MPNST is composed of plump, epithelioid cells with abundant eosinophilic cytoplasm [121]. MPNSTs could show an enhanced level of pleomorphism; in such cases, the differentials include a variety of pleomorphic sarcomas, particularly DDLPS [49]. Recently, the term “atypical neurofibromatous neoplasms of uncertain biologic potential” was proposed for lesions displaying at least two of the following four features in patients with NF1: (1) cytological atypia, (2) loss of neurofibroma architecture, (3) hypercellularity, and (4) mitotic index >1/50 and <3/10 HPFs [122]. Immunohistochemically, tumor cells are focally positive for S100 protein (<50% of cases), SOX10 (<70% of cases), and GFAP (20–30% of cases) [108]. Complete loss of H3K27me3 is helpful in the diagnosis of MPNST, with high-grade tumors showing more frequent loss than low-grade tumors [123].

Differential diagnoses of MPNST include schwannoma, DDLPS, leiomyosarcoma, RMS, synovial sarcoma, and malignant melanoma. Schwannomas lack malignant cytologic atypia and are strongly and diffusely positive for S100 protein. DDLPSs show components of WDLPS and are positive for MDM2 and CDK4. Leiomyosarcomas show smooth muscle cytomorphology and are positive for SMA, caldesmon, and desmin. RMSs do not arise in association with a large nerve or within the context of NF1 and show retained (i.e., normal) nuclear staining for H3K27me3 [124]. Monophasic synovial sarcomas show more uniform nuclei with wiry stromal collagen and are diffusely positive for TLE1 and SS18-SSX and focally positive for cytokeratin and EMA. Primary or metastatic retroperitoneal malignant melanomas are extremely rare and show spindle and epithelioid cells with severe nuclear atypia and are diffusely positive for S100 protein and other melanocytic markers (e.g., HMB-45, melan-A).

### 4.5. Extraskeletal Osteosarcoma

Extraskeletal osteosarcoma (EOS) is a malignant tumor characterized by production of osteoid or bone matrix by neoplastic cells and arises without connection to the skeletal system [125]. EOS accounts for <1% of all soft tissue sarcomas and around 4% of all osteosarcomas. It usually arises in midlife and late adulthood. The majority of cases develop de novo. Approximately 5–10% of cases develop at sites of previous irradiation [126]. EOS most commonly occurs in the limbs, especially the thigh, but the anatomic distribution is wide [127]. The retroperitoneum is a frequent site for EOS [128]. Plain radiographs, CT, and MRI usually reveal a large soft tissue mass with variable calcification.

EOS shows a broad spectrum of histologic patterns. All the major types of osteosarcoma that arise in bone are seen in EOS. Depending on the dominant histologic pattern, osteosarcomas are divided into osteoblastic, fibroblastic, chondroblastic, telangiectatic, and small cell types (Figure 9). Osteoblastic type is the most common pattern. Mixed patterns are frequently present. The tumor cells are variably pleomorphic spindle or polygonal cells. Abundant mitotic activity with atypical mitotic figures is present. The presence of neoplastic osteoid (unmineralized matrix) and bone is necessary for diagnosis. Neoplastic bone is intimately associated with malignant tumor cells and varies from thin lace-like, trabeculae to compact bone. Amplification of *MDM2* can be detected in some of the high-grade EOSs [129]. Immunohistochemically, SATB2 is sensitive but not specific for osteosarcoma [130,131].

Differential diagnoses of EOS include benign and malignant bone-forming lesions such as myositis ossificans, DDLPS, MPNST, ossifying synovial sarcoma, and UPS. Myositis ossificans shows zonal distribution with peripheral bone maturation and lacks malignant cytologic atypia [132]. DDLPSs with heterologous osteosarcomatous components should be distinguished from EOS [133]. DDLPSs show components of WDLPS and are positive for MDM2 and CDK4. MPNSTs exhibit hyperchromatic spindle cells in fascicles and are focally positive for S100 protein. Ossifying synovial sarcomas demonstrate uniform spindle tumor cells and are positive for cytokeratin, EMA, and TLE1 and reveal t(X;18) with *SSX2* involvement [134]. UPSs have significant morphological overlap with EOSs but do not show neoplastic bone formation. Careful sampling of the lesions and exclusion of the skeletal origin are required for a correct diagnosis of EOS.

### 4.6. Synovial Sarcoma

Synovial sarcoma is a monomorphic blue spindle cell sarcoma showing variable epithelial differentiation [135]. Synovial sarcoma is characterized by a specific chromosomal translocation t(X;18)(p11;q11) with *SS18-SSX* fusion gene. It may occur at any age. More than half of the cases occur in adolescents or young adults [136]. The majority (70%) of synovial sarcomas arise in the deep soft tissue of the lower and upper extremities, often in juxta-articular locations. It may arise primarily in a wide variety of visceral locations, such as the gastrointestinal tract and kidneys [137,138]. Synovial sarcoma rarely develops in the pelvis and retroperitoneum [139]. Up to one third of synovial sarcomas have radiologically detectable calcification that is occasionally extensive.

Histologically, synovial sarcoma is classified into monophasic (spindle cell), biphasic, and poorly differentiated types. Monophasic type is the most common. Monophasic synovial sarcomas are composed of monomorphic spindle cells with granular chromatin, inconspicuous nucleoli, and poorly defined cytoplasm. Stromal mast cells and a branching, hemangiopericytoma-like vascular pattern are present in a variable amount of collagenous stroma. Biphasic synovial sarcoma has epithelial and spindle cell components in varying proportions. Poorly differentiated synovial sarcoma accounts for approximately 5–10% of cases and is characterized by increased cellularity, greater nuclear atypia, and high mitotic activity (>6 mitoses/mm^2^ or >10 mitoses per 10 HPFs of 0.17 mm^2^) (Figure 10) [135,140]. Immunohistochemically, spindle tumor cells are focally positive for EMA and cytokeratin. Strong and diffuse nuclear staining for TLE1 is present [141]. Although TLE1 is diagnostically useful, it is not totally specific for synovial sarcoma. Recently, a novel SS18-SSX fusion antibody for the diagnosis of synovial sarcoma was described [67].

Differential diagnoses of synovial sarcoma include SFT, MPNST, leiomyosarcoma, and extraskeletal ES. SFTs show a “patternless pattern”, with branching and staghorn vascular patterns, and are diffusely positive for CD34 and STAT6. MPNSTs have wavy, buckled nuclei and are variably positive for S100 protein with loss of H3K27me3 expression. Leiomyosarcomas have blunt-ended, cigar-shaped nuclei, brightly eosinophilic cytoplasm, and are positive for SMA, caldesmon, and desmin. Extraskeletal ESs show diffuse membranous positivity for CD99, nuclear positivity for FLI1, and diffuse nuclear expression of NKX2.2. Detection of *SS18-SSX* fusion transcripts is diagnostic and particularly valuable in synovial sarcomas arising in atypical locations outside the extremities [142].

### 4.7. Desmoplastic Small Round Cell Tumor

Desmoplastic small round cell tumor (DSRCT) is a malignant mesenchymal neoplasm composed of small round tumor cells with polyphenotypic differentiation and prominent stromal desmoplasia [143]. DSRCT was first described by Gerald and Rosai in 1989 [144]. It is characterized by a recurrent chromosomal translocation t(11;22)(p13;q12), leading to *EWSR1-WT1* gene fusion. DSRCT primarily affects children and young adults, with a male predominance. It most commonly arises in the abdominal cavity. It frequently involves any part of the peritoneal cavity and retroperitoneum [145,146]. Multiple serosal implants are common. Despite the use of multimodality therapy, it is a highly aggressive neoplasm with extremely poor prognosis [147]. Clinically, patients present with abdominal distention, palpable masses, ascites, and organ obstruction.

Histologically, DSRCT shows nests of small round tumor cells in the desmoplastic stroma. The tumor cells have uniform, small hyperchromatic nuclei, scant cytoplasm, and indistinct cytoplasmic borders (Figure 11). One third of DSRCT cases exhibit a wide range of morphological features. Rhabdoid appearance, epithelial features with gland formation, a large cell variant, spindle-shaped morphology, and rosette formation may be seen [148,149]. Desmoplastic stroma is composed of fibroblasts and myofibroblasts in a collagenous matrix. A variably prominent stromal vascularity is present. Immunohistochemically, the tumor cells show a distinct immunophenotype of polyphenotypic differentiation, including expression of epithelial (e.g., cytokeratin, EMA), muscular (e.g., desmin), and neural markers (e.g., NSE, Leu-7). Desmin immunoreactivity for desmin typically appears as a perinuclear dot-like pattern. Nuclear expression of WT1 is present in the vast majority of cases. WT1 immunoreactivity is a useful marker to differentiate DSRCT from other small round cell tumors [150].

Differential diagnoses of DSRCT include extraskeletal ES, alveolar RMS, extrarenal rhabdoid tumor, neuroblastoma, and metastatic neuroendocrine carcinoma. Extraskeletal ES lacks prominent desmoplastic stroma and shows diffuse membranous positivity for CD99 and nuclear expression of NKX2.2. Alveolar RMSs show nests of central discohesion, with a lack of prominent desmoplastic stroma and positive myogenin and MYOD1 immunoreactivity. Extrarenal rhabdoid tumors show sheets of epithelioid cells and are negative for desmin with loss of nuclear INI1 expression. Neuroblastomas exhibit lobular architecture, an eosinophilic neurofibrillary matrix, a lack of *EWSR1* translocation, and negative cytokeratin and desmin immunoreactivity. PHOX2B is a sensitive and specific marker for neuroblastoma [151]. Metastatic neuroendocrine carcinomas have a distinctive salt and pepper chromatin pattern and are negative for desmin and WT1.

### 4.8. PEComa

Perivascular epithelioid cell tumors (PEComas) are mesenchymal neoplasms composed of perivascular epithelioid cells (distinctive epithelioid cells that are often closely associated with blood vessel walls) and express both melanocytic and smooth muscle markers [152]. PEComas show a wide anatomical distribution and usually occur in young to middle aged adults, with a marked female predominance. Most often, these tumors arise in the retroperitoneum, abdominopelvic region, gastrointestinal tract, and uterus [153,154]. Most PEComas are sporadic; a small subset is associated with tuberous sclerosis complex. Deletion of the *TSC2* gene on chromosome 16p and its consequent mTOR activation play a relevant role in the neoplastic process [28]. Clinically, PEComas usually present as a painless mass.

Histologically, PEComa shows epithelioid cells with vesicular nuclei and abundant granular eosinophilic or clear cytoplasm. The tumor cells are arranged in a nested and sheet-like pattern, with surrounding thin-walled, branching blood vessels. Spindle-shaped cells resembling smooth muscle cells and very pleomorphic cells can also be seen. Melanin pigment is occasionally present. Morphological variants include sclerosing PEComa and fibroma-like PEComa [155,156]. Sclerosing PEComas frequently arise in the retroperitoneum and show a densely collagenous stroma (Figure 12). The provisional classification of PEComas into benign, uncertain malignant potential and malignant categories was proposed by Folpe et al. [157]. Lesions with two or more worrisome features (tumor size >5cm, infiltrative growth, high nuclear grade, high cellularity, necrosis, mitotic activity >1 mitotic figure/50 HPFs, vascular invasion) are classified as malignant PEComa. Immunohistochemically, the tumor cells are positive for melanocytic markers (e.g., HMB-45, melan-A) and smooth muscle markers (e.g., SMA, desmin, caldesmon). TFE3 is positive in a distinct subset of PEComas that harbor *TFE3* gene fusions [158].

Differential diagnoses of PEComa include GIST, leiomyosarcoma, alveolar soft part sarcoma, malignant melanoma, and metastatic clear cell renal cell carcinoma. GISTs show uniform spindle cells with eosinophilic cytoplasm and indistinct cell borders and are positive for CD34, CD117, and DOG1. Leiomyosarcomas have fascicles of spindle cells with blunt-ended nuclei and brightly eosinophilic cytoplasm. Additionally, they lack a delicate capillary network and are negative for melanocytic markers. Alveolar soft part sarcomas show an alveolar pattern of large eosinophilic or clear epithelioid cells and are positive for TFE3 and negative for melanocytic markers. Malignant melanomas show markedly cytologic atypia and are strongly positive for S100 protein and negative for smooth muscle markers. Metastatic clear cell renal cell carcinomas are positive for EMA and PAX8 and negative for melanocytic markers.

### 4.9. Undifferentiated Pleomorphic Sarcoma

Undifferentiated soft tissue sarcoma (USTS) shows no identifiable line of differentiation when analyzed with available technology [159]. At present, it is a heterogeneous group and a diagnosis of exclusion. It accounts for as many as 20% of all soft tissue sarcomas. USTSs are broadly divided into pleomorphic, spindle cell, round cell, and epithelioid groups [160]. UPS (classified as pleomorphic malignant fibrous histiocytoma in the past) represents the largest group and occurs mostly in older adults. UPSs rarely arise in the retroperitoneum. The etiology of most USTSs is unknown. A subset of cases is radiation-associated [161]. UPSs typically show a nonspecific complex karyotype with numerous genomic rearrangements. Clinically, UPSs have no characteristic clinical features.

Histologically, UPS is composed of highly atypical pleomorphic and/or spindle cells arranged in a storiform, fascicular, or patternless arrangement. Bizarre multinucleated tumor giant cells are frequently present. Abundant mitotic activity, often including atypical forms, is also observed. Pleomorphic soft tissue sarcomas with myogenic differentiation are significantly more aggressive [162,163]. Immunohistochemically, UPSs often show a small number of cells that express SMA, CD34, and cytokeratin. However, these findings are non-specific. Immunohistochemistry and molecular genetics play a major role in excluding other diagnoses.

Differential diagnoses of UPS are broad and include DDLPS, pleomorphic leiomyosarcoma, pleomorphic MPNST, pleomorphic RMS, malignant melanoma, and metastatic sarcomatoid carcinoma. DDLPSs have components of WDLPS and are positive for MDM2 and CDK4 and *MDM2* amplification. Most cases of retroperitoneal sarcomas initially diagnosed as so-called malignant fibrous histiocytomas are DDLPS [50,164]. Pleomorphic leiomyosarcomas show at least focally eosinophilic spindle cells with blunt-ended nuclei. These cells have a fascicular pattern and are diffusely positive for SMA and variably positive for desmin. In pleomorphic MPNST, the tumor cells are focally positive for S100 protein and SOX10. Complete loss of H3K27me3 is helpful in the diagnosis of MPNST. In pleomorphic RMS, the tumor cells are positive for desmin, myogenin, and MYOD1. Malignant melanomas are positive for S100 protein and melanocytic markers. Metastatic sarcomatoid carcinomas can be excluded by clinical history and are positive for epithelial markers. In cases of pleomorphic malignant neoplasms developing in the retroperitoneum, pathologists should perform extensive sampling combined with appropriate immunohistochemical panels and molecular testing to arrive at a correct diagnosis.

### 4.10. Extraskeletal Ewing Sarcoma

Ewing sarcoma (ES) is a small round cell sarcoma showing gene fusions involving female expressed transcript (FET) family genes (usually *EWSR1*) and the erythroblast transformation-specific (ETS) family of transcription factors [165]. ES is the second most common sarcoma of the bone in children and young adults, after osteosarcoma. Around 10–20% of cases are extraskeletal. Extraskeletal ES is most common between the ages of 10 and 30 years. It has a wide anatomical distribution and commonly arises in the extremities (thigh) and trunk (paravertebral region) [166,167]. A variety of visceral locations have been reported [168,169]. Retroperitoneal extraskeletal ESs are rare [170,171]. Although the majority of cases are sporadic, germline mutations have been detected in around 10% of cases [172].

Histologically, ES is composed of uniform, small round tumor cells. The tumor cells have finely dispersed chromatin with inconspicuous or small nucleoli and scanty, clear, or eosinophilic cytoplasm (Figure 13) [173]. The tumor cells are arranged in a lobular or trabecular pattern. Necrosis is frequently found. Neuroectodermal differentiation (Homer Wright rosettes) may be observed. Rare morphological variants include atypical (large cell) and adamantinoma-like variants [174,175]. Atypical EW consists of rather large tumor cells with conspicuous nucleoli and irregular contours. Adamantinoma-like ES shows strong cytokeratin expression or focal keratinization. Immunohistochemically, the tumor cells show a strong and diffuse membranous expression of CD99. Nuclear NKX2.2 expression is more specific for ES than CD99 [176]. Strong nuclear ERG immunoreactivity is specific for ES with *EWSR1-ERG* rearrangement [177]. Cyclin D1 can be exploitable as a diagnostic adjunct to conventional markers in confirming the diagnosis of ES [178].

Differential diagnoses of extraskeletal ES include alveolar RMS, poorly differentiated synovial sarcoma, DSRCT, *CIC*-rearranged sarcoma, and sarcoma with *BCOR* genetic alterations. Alveolar RMSs show an alveolar pattern and are positive for desmin and myogenin. Poorly differentiated synovial sarcomas show strong and diffuse positivity for TLE1 and *SS18* gene fusions. DSRCTs exhibit a desmoplastic stroma and are positive for cytokeratin, desmin, NSE, and WT1. Recently defined *CIC*-rearranged sarcomas and sarcomas with *BCOR* genetic alterations may occur in the retroperitoneum. *CIC*-rearranged sarcomas show diffuse sheets of undifferentiated small round cells and strong nuclear expression of WT1 and ETV [179]. Sarcomas with *BCOR* genetic alterations show solid sheets of uniform round to ovoid cells in a myxoid matrix and strong nuclear expression of BCOR and CCNB3 [180].

## 5. Miscellaneous Retroperitoneal Mesenchymal Tumors

Myxofibrosarcoma [181], alveolar soft part sarcoma [182], angiosarcoma [183], clear cell sarcoma of soft tissue [184], and extraskeletal myxoid chondrosarcoma [185] rarely occur in the retroperitoneum. Each of these sarcomas is characterized by distinctive clinicopathologic features and unique genetic findings. Benign mesenchymal tumors, including lipoma [37], leiomyoma [186], schwannoma, and neurofibroma, occur in the retroperitoneum. These tumors may mimic sarcoma. It is important to exclude benign mesenchymal tumors, malignant melanomas, germ cell tumors, malignant lymphomas, parenchymal tumors of retroperitoneal organs (e.g., pancreas, kidneys) located in the retroperitoneal space, and metastatic carcinomas before diagnosing a primary retroperitoneal sarcoma.

## 6. Conclusions

Retroperitoneal sarcomas constitute a rare and diagnostically challenging group of sarcomas that show a wide range of differentiation. Herein, we provide a practical diagnostic approach to retroperitoneal sarcomas and review their histologic features. Adequate sampling for histologic evaluation is necessary. Furthermore, careful integration of clinical history, imaging findings, histopathology, immunohistochemistry, and molecular testing is important for correct diagnosis and management. Recent advances in molecular genetic alterations and the emergence of novel diagnostic immunohistochemical markers may further improve diagnostic accuracy for soft tissue tumors.

## Figures and Tables

**Figure 1 diagnostics-10-00642-f001:**
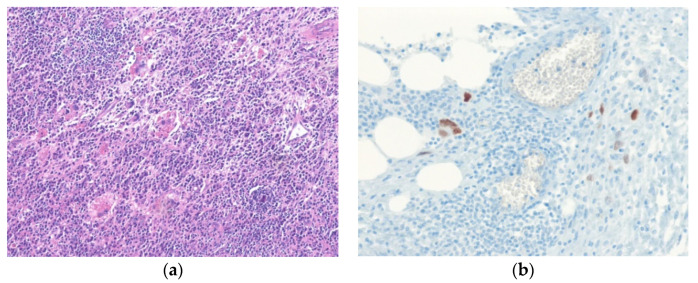
Well-differentiated liposarcoma, inflammatory subtype. (**a**) The tumor shows an abundant chronic inflammatory cell infiltrate and scattered atypical pleomorphic stromal cells. (**b**) The atypical stromal cells are positive for MDM2 (H&E stain, original magnification 100× **a**; MDM2 immunostain, original magnification 200× **b**).

**Figure 2 diagnostics-10-00642-f002:**
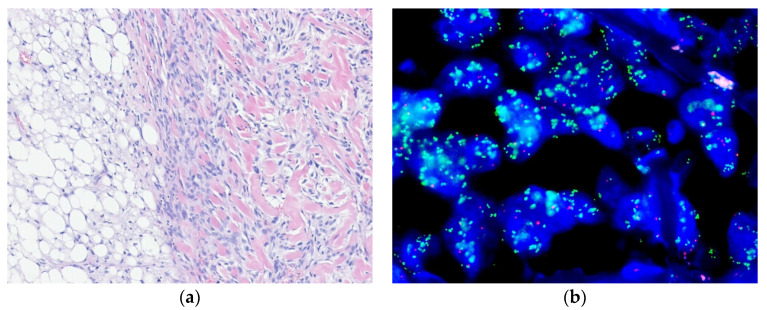
Dedifferentiated liposarcoma. (**a**) The tumor shows an abrupt transition from well-differentiated liposarcoma component to a non-lipogenic, high-grade dedifferentiated area (right). (**b**) The dedifferentiated component shows *MDM2* amplification by FISH. The green signal corresponds to the *MDM2* probe, while the red signal corresponds to the chromosome 12 centromeric probe (H&E stain, original magnification 100× **a**; *MDM2* FISH, original magnification 1000× **b**).

**Figure 3 diagnostics-10-00642-f003:**
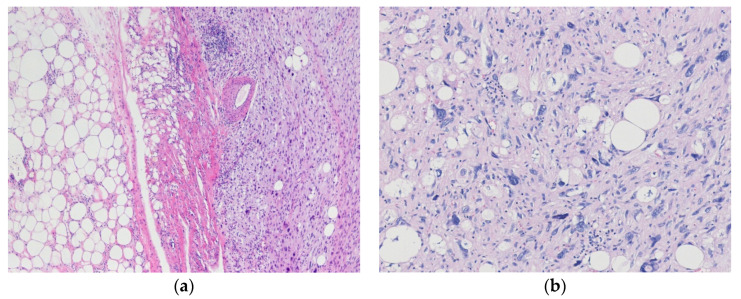
Dedifferentiated liposarcoma with homologous lipoblastic differentiation. (**a**) The tumor shows an abrupt transition from well-differentiated liposarcoma component to dedifferentiated component. (**b**) The dedifferentiated component shows homologous lipoblastic differentiation with scattered lipoblasts. The morphology resembles pleomorphic liposarcoma (H&E stain, original magnifications 100× **a** and 200× **b**).

**Figure 4 diagnostics-10-00642-f004:**
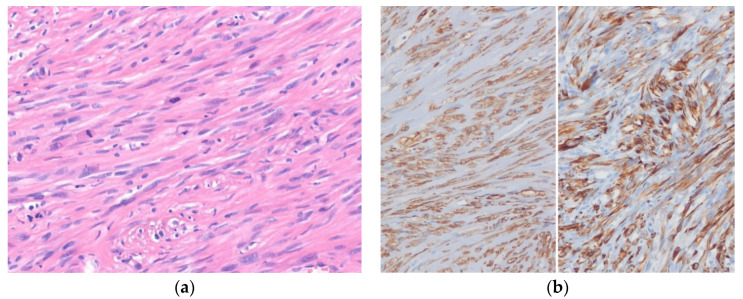
Leiomyosarcoma. (**a**) The tumor cells have cigar-shaped, blunt-ended nuclei with brightly eosinophilic cytoplasm and a fascicular pattern. Mitoses are present. (**b**) The tumor cells are diffusely positive for SMA (left) and desmin (right) (H&E stain, original magnification 20 × **a**; SMA and desmin immunostain, original magnification 200× **b**).

**Figure 5 diagnostics-10-00642-f005:**
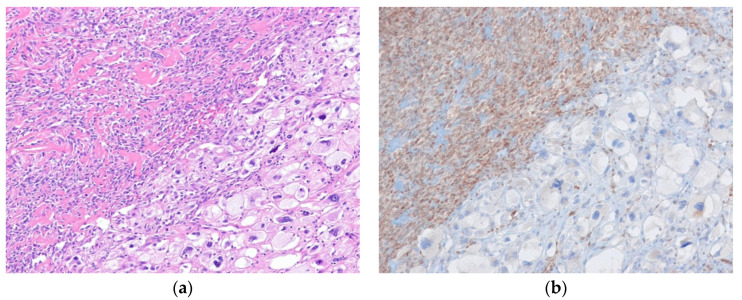
Dedifferentiated solitary fibrous tumor. (**a**) The tumor shows an abrupt transition from typical solitary fibrous tumor to high-grade dedifferentiated area. (**b**) The typical solitary fibrous area shows strong and diffuse positive nuclear expression of STAT6 and the dedifferentiated area shows loss of STAT6 expression (H&E stain, original magnification 100× **a**; STAT6 immunostain, original magnification 100× **b**).

**Figure 6 diagnostics-10-00642-f006:**
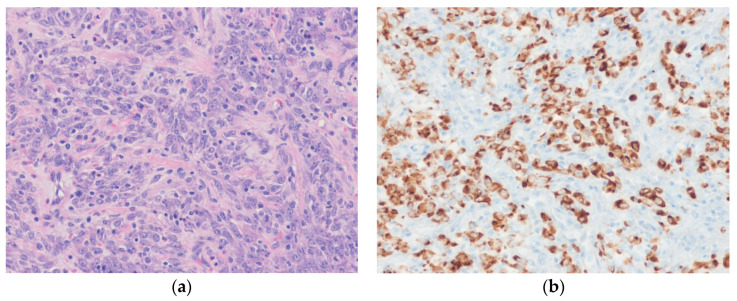
Epithelioid inflammatory myofibroblastic sarcoma. (**a**) The tumor cells are round to epithelioid-shaped and have prominent nucleoli, with mild infiltration of inflammatory cells, including neutrophils and lymphocytes. (**b**) The tumor cells show nuclear membranous staining for ALK (H&E stain, original magnification 200× **a**; ALK immunostain, original magnification 200× **b**).

**Figure 7 diagnostics-10-00642-f007:**
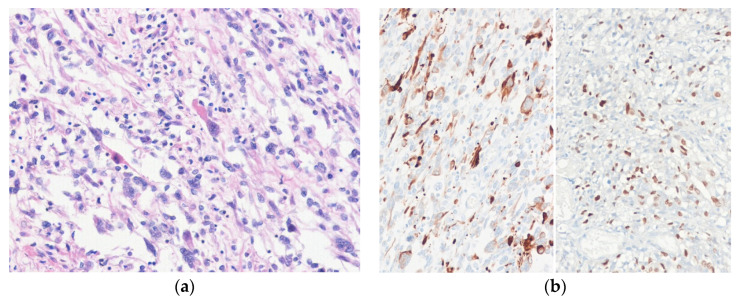
Embryonal rhabdomyosarcoma. (**a**) The tumor is composed of primitive round to spindle cells and eosinophilic rhabdomyoblasts. (**b**) The tumor cells are positive for desmin (left) and myogenin (right) (H&E stain, original magnification 200× **a**; desmin and myogenin immunostain, original magnification 200× **b**).

**Figure 8 diagnostics-10-00642-f008:**
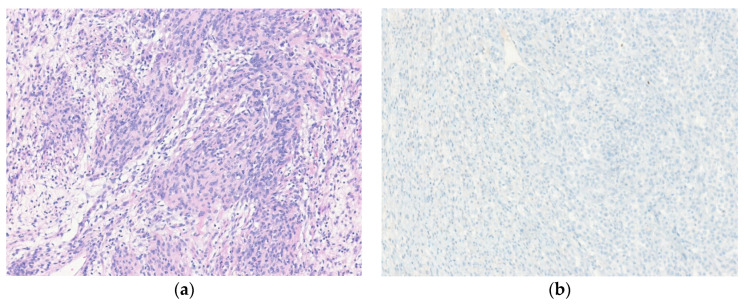
Malignant peripheral nerve sheath tumor. (**a**) The tumor is composed of spindle cells with alternating hypercellular and hypocellular areas. (**b**) The tumor cells show loss of H3K27me3 expression (H&E stain, original magnification 100× **a**; H3K27me3 immunostain, original magnification 200× **b**).

**Figure 9 diagnostics-10-00642-f009:**
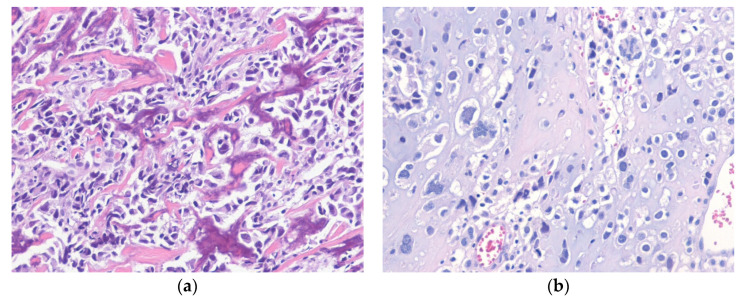
Extraskeletal osteosarcoma. (**a**) The tumor consists of hyperchromatic tumor cells producing lace-like neoplastic bone. (**b**) The tumor shows a cartilaginous area (H&E stain, original magnification 200× **a** and 200× **b**).

**Figure 10 diagnostics-10-00642-f010:**
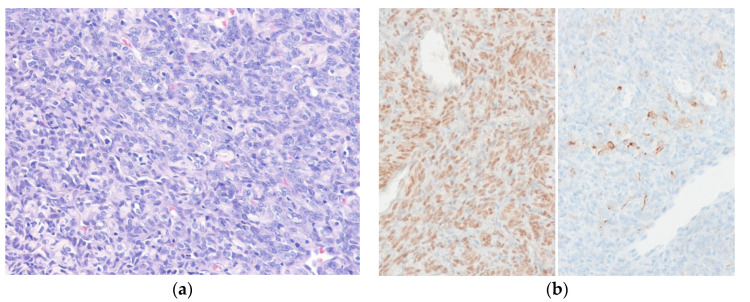
Poorly differentiated synovial sarcoma. (**a**) The tumor shows hypercellular rounded tumor cells. Slightly spindled tumor cells are also present. Mitotic figures are frequently seen. (**b**) The tumor cells are strongly and diffusely positive for TLE1 (left) and focally positive for cytokeratin (AE1/AE3) (right) (H&E stain, original magnification 200× **a**; TLE1 and cytokeratin (AE1/AE3) immunostain, original magnification 200× **b**).

**Figure 11 diagnostics-10-00642-f011:**
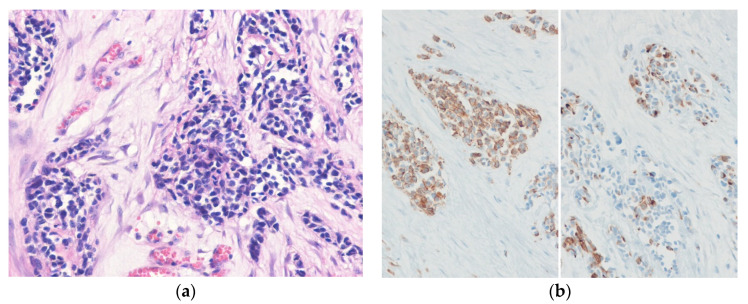
Desmoplastic small round cell tumor. (**a**) The tumor shows nests of small round tumor cells within desmoplastic fibrous stroma. (**b**) The tumor cells are positive for cytokeratin (AE1/AE3) (left) and show perinuclear dot-like expression for desmin (right) (H&E stain, original magnification 200× **a**; cytokeratin (AE1/AE3) and desmin immunostain, original magnification 200× **b**).

**Figure 12 diagnostics-10-00642-f012:**
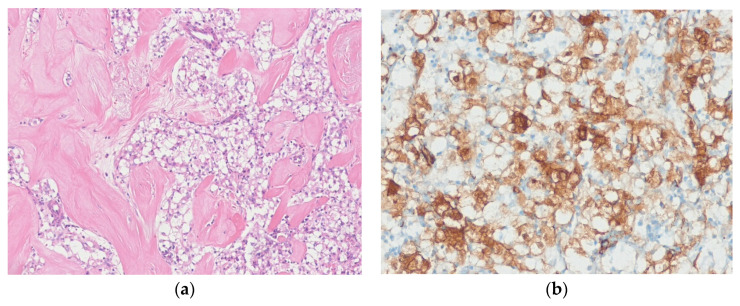
Sclerosing PEComa. (**a**) The tumor shows nests of epithelioid tumor cells with abundant clear cytoplasm. The densely hyalinized collagenous stroma is present. (**b**) The tumor cells are positive for SMA (H&E stain, original magnification 100× **a**; SMA immunostain, original magnification 200× **b**).

**Figure 13 diagnostics-10-00642-f013:**
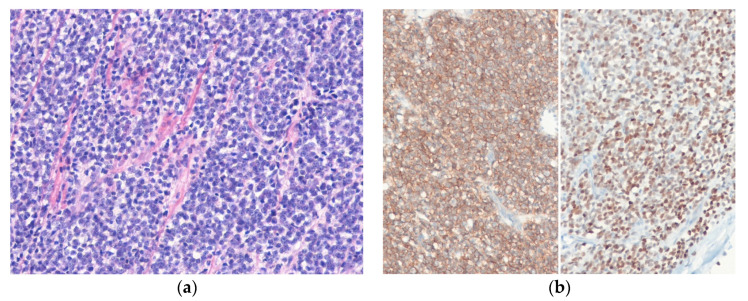
Extraskeletal Ewing sarcoma. (**a**) The tumor is composed of uniform small round tumor cells arranged in vaguely lobular pattern. The tumor cells have finely dispersed chromatin and scanty cytoplasm. (**b**) The tumor cells show diffusely membranous expression for CD99 (left) and diffusely nuclear expression for NKX2.2 (right) (H&E stain, original magnification 100× **a**; CD99 and NKX2.2 immunostain, original magnification 100× **b**).

**Table 1 diagnostics-10-00642-t001:** Relatively common and rare sarcomas arising in the retroperitoneum.

Relatively Common Sarcomas	Rare Sarcomas
Liposarcoma (well-differentiated and dedifferentiated subtypes) Leiomyosarcoma	Solitary fibrous tumor (malignant) Inflammatory myofibroblastic tumor Rhabdomyosarcoma MPNST Extraskeletal osteosarcoma Synovial sarcoma Desmoplastic small round cell tumor PEComa (malignant) Undifferentiated pleomorphic sarcoma Extraskeletal Ewing sarcoma

MPNST, malignant peripheral nerve sheath tumor; PEComa, perivascular epithelioid cell tumor.

**Table 2 diagnostics-10-00642-t002:** Pattern and additional findings of retroperitoneal mesenchymal tumors.

Pattern	Tumor Types
Spindle cell	Leiomyosarcoma Solitary fibrous tumor Inflammatory myofibroblastic tumor MPNST Monophasic synovial sarcoma
Round cell	Poorly differentiated synovial sarcoma Desmoplastic small round cell tumor Extraskeletal Ewing sarcoma
Epithelioid cell	Epithelioid inflammatory myofibroblastic sarcoma Epithelioid MPNST PEComa
Pleomorphic cell	Dedifferentiated liposarcoma Pleomorphic leiomyosarcoma Pleomorphic rhabdomyosarcoma Undifferentiated pleomorphic sarcoma
Adipocytic component	Liposarcoma (well differentiated, dedifferentiated, myxoid, and pleomorphic subtype) Lipomatous (fat-forming) solitary fibrous tumor
Prominent inflammatory cells	Inflammatory well-differentiated liposarcoma Inflammatory myofibroblastic tumor
Tumor osteoid and bone	Extraskeletal osteosarcoma

MPNST, malignant peripheral nerve sheath tumor; PEComa, perivascular epithelioid cell tumor.

**Table 3 diagnostics-10-00642-t003:** Immunohistochemistry of selected retroperitoneal mesenchymal tumors.

	Liposarcoma (Well-Differentiated/Dedifferentiated)	Solitary Fibrous Tumor	Inflammatory Myofibroblastic Tumor	Leiomyosarcoma	MPNST	Synovial Sarcoma	DSRCT	PEComa	Extraskeletal Ewing Sarcoma
SMA	− ^(a)^	+/−	+/−	+	−	−	−	+	−
Desmin	− ^(a)^	−	+/−	+	−	−	+	+/−	−
CD34	−	+	+/−	−	+	−	−	−	−
S100 protein	+/−	−	−	−	+	+/−	−	+/−	+/−
MDM2	+	−	−	−	− ^(e)^	−	−	−	−
CDK4	+	−	−	−	−	−	−	−	−
STAT6	− ^(b)^	+ ^(d)^	−	−	−	−	−	−	−
ALK	−	−	+	−	− ^(f)^	−	−	−	−
CD99	−	+	−	−	−	+	+	−	+
H3K27me3	Retained ^(c)^	Retained	NA	Retained	Loss ^(g)^	Retained ^(h)^	NA	NA	Retained
NKX2.2	−	−	−	−	−	− ^(i)^	− ^(j)^	−	+
TLE1	−	+/−	−	−	+/−	+	−	−	−
SOX10	−	−	−	−	+	−	−	−	−
WT1	−	−	−	−	−	−	+	−	−
HMB-45, melan-A	−	−	−	−	−	−	−	+	−
Cytokeratin, EMA	−	+/−	+/−	+/−	−	+	+	−	+/−

MPNST, malignant peripheral nerve sheath tumor; DSRCT, desmoplastic small round cell tumor; PEComa, perivascular epithelioid cell tumor; +, positive staining; +/−, focal or variable staining; −, negative staining; NA, no available data; ^(a)^ can be expressed in well-differentiated and dedifferentiated liposarcoma, ^(b)^ may be expressed in a subset of dedifferentiated liposarcoma, ^(^^c)^ may be lost in a subset of dedifferentiated liposarcoma, ^(^^d)^ may be decreased or lost in dedifferentiated solitary fibrous tumor, ^(^^e)^ can be overexpressed in a subset of MPNST, ^(f)^ may be expressed in a subset of MPNST, ^(^^g)^ lost in approximately 50% of MPNSTs (in 90% of high-grade MPNSTs), ^(^^h)^ may be lost in a subset of synovial sarcoma, ^(i)^ may be expressed in a subset of poorly differentiated synovial sarcoma, ^(^^j)^ may be expressed in a subset of DSRCT.

**Table 4 diagnostics-10-00642-t004:** Cytogenetic and molecular alterations in selected retroperitoneal mesenchymal tumors.

Tumor Types	Cytogenetic Alterations	Molecular Alterations
Well-differentiated liposarcoma	Supernumerary ring or giant marker chromosome(s)	*MDM2* amplification, other co-amplified genes *CDK4*, *HMGA2*, *TSPAN31*, *YEATSA4*, *CPM*, *FRS2*
Dedifferentiated liposarcoma	Supernumerary ring or giant marker chromosome(s)	*MDM2* amplification, other co-amplified genes *CDK4*, *JUN*, *TERT*, *CPM*, *MAP3K5*
Myxoid liposarcoma	t(12;16)(q13;p11) t(12;22)(q13;q12)	*FUS-DDIT3* *EWSR1-DDIT3*
Solitary fibrous tumor	Inv(12)(q13q13)	*NAB2-STAT6* fusion
Inflammatory myofibroblastic tumor	t(1;2)(q22;p23) t(2;19)(p23;p13) t(2;17)(p23;q23)	*TPM3-ALK* fusion *TPM4-ALK* fusion *CLTC-ALK* fusion *ROS1* and *PDGFRB* rearrangement
Epithelioid inflammatory myofibroblastic sarcoma	t(2;2)(p23;q13)	*RANBP2-ALK* fusion
Embryonal rhabdomyosarcoma	Complex karyotypes; loss of heterozygosity at 11p15.5	
Alveolar rhabdomyosarcoma	t(2;13)(q35;q14) t(1;13)(p36;q14)	*PAX3-FOXO1A* fusion *PAX7-FOXO1A* fusion
MPNST	Complex karyotypes; inactivation mutations in *NF1*, *CDKN2A*/*CDKN2B*, *EED,* or *SUZ12*	
Epithelioid MPNST	*SMARCB1* gene inactivation	
Synovial sarcoma	t(X;18)(p11;q11)	*SS18-SSX1* fusion *SS18-SSX2* fusion
Desmoplastic small round cell tumor	t(11;22)(p13;q12)	*EWSR1-WT1*, *EWSR1-ERG, EWDR1-FLI1* fusion
PEComa	Deletion of 16p, the location of *TSC2* gene	*SFPQ-TFE3*, *DVL2-TFE3*, *NONO-TFE3* fusion
Extraskeletal Ewing sarcoma	t(11;22)(q24;q12) t(21;22)(q12;q12) t(2;22)(q33;q12) t(7;22)(p22;q12) t(17;22)(q12;q12)	*EWSR1-FLI1* fusion *EWSR1-ERG* fusion *EWSR1-FEV* fusion *EWSR1-ETV1* fusion *EWSR1-E1AF* fusion

MPNST, malignant peripheral nerve sheath tumor; PEComa, perivascular epithelioid cell tumor.

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
