# Peer review of "Retroperitoneal Sarcomas: An Update on the Diagnostic Pathology Approach"

_diagnostics, 2020, doi:10.3390/diagnostics10090642_

Round 1
Reviewer 1 Report
This is a well written review and update on retroperitoneal sarcomas and the diagnostic approach.
I have some minor comments/remarks:
*page 5 (Table 3): well-differentiated and dedifferentiated liposarcoma can show myogenic differentiation (Gronchi A, et al. 2015;39(3):383-393).
*page 5 (Table 3): MDM2 overexpression can be observed in MPNST (Makise N, et al. Am J Surg Pathol 2018)
*page 5 (Table 3) CDK4 (in stead of DCK4)
*line 34: PS?
*Recently, a novel SS18-SSX fusion antibody for the diagnosis of synovial sarcoma is described in literature (Baranov E, et al. Am J Surg Pathol 2020;44(7):922-933.
p15 line 351....for TLE1 and SS18-SSX; line 160 ...are diffusely positive for TLE1 and SS18-SSX; line 198...diffusely positive for TLE1 and SS18-SSX
*line 100-101: MPNST's show fascicles of spindle cells with wavy, tapering nuclei and are negative for MDM2 and CDK4-->MDM2 expression can be seen in MPNST, see above
*p12, line 231 ...for S100 and SOX10 in <50% and <70% of cases respectively
*p18: PHOX2B as sensitive and specifica marker for neuroblastoma (Hung YP, et al. Histopathology 2017;71(5):786-794)
*p15, line 327: liposarcomatous differerentiation is also described as a rare phenomenon in MPNST (Van Haverbeke C, et al. Pathology 2018;50(4)475-478)
*STAT6 expression in dedifferentiated liposarcoma: add references
Creytens D, et al. Appl Immunohistochem Mol Morphol 2015;23(6):462-463
Creytens D. Pathology 2017;49(6):679
*rhabdomyoblastic differentiation is also described as a rare phenomenon in malignant SFT
Creytens D, et al. Int J Surg Pathol 2018;26(5):423-427
Author Response
This is a well written review and update on retroperitoneal sarcomas and the diagnostic approach.
I have some minor comments/remarks:
*page 5 (Table 3): well-differentiated and dedifferentiated liposarcoma can show myogenic differentiation (Gronchi A, et al. 2015;39(3):383-393).
-> WDLPS and DDLPS can show myogenic differentiation. (in text)
a) Can be expressed in well-differentiated and dedifferentiated liposarcoma. (in Table 3)
*page 5 (Table 3): MDM2 overexpression can be observed in MPNST (Makise N, et al. Am J Surg Pathol 2018)
-> MDM2 overexpression can be observed in MPNST. (in text)
e) Can be overexpressed in a subset of MPNST. (in Table 3)
*page 5 (Table 3) CDK4 (in stead of DCK4)
-> CDK4
*line 34: PS?
-> WDLPS
*Recently, a novel SS18-SSX fusion antibody for the diagnosis of synovial sarcoma is described in literature (Baranov E, et al. Am J Surg Pathol 2020;44(7):922-933.
-> ‘Recently, a novel SS18-SSX fusion antibody for the diagnosis of synovial sarcoma is described.’
* p15 line 351....for TLE1 and SS18-SSX; line 160 ...are diffusely positive for TLE1 and SS18-SSX; line 198...diffusely positive for TLE1 and SS18-SSX
-> diffusely positive for TLE1 and SS18-SSX
*line 100-101: MPNST's show fascicles of spindle cells with wavy, tapering nuclei and are negative for MDM2 and CDK4-->MDM2 expression can be seen in MPNST, see above
-> MDM2 expression can be seen in MPNST. High-level MDM2 amplification strongly suggests DDLPS over MPNST.
*p12, line 231 ...for S100 and SOX10 in <50% and <70% of cases respectively
-> S100 protein and SOX10 in <50% and <70% of cases, respectively.
*p18: PHOX2B as sensitive and specifica marker for neuroblastoma (Hung YP, et al. Histopathology 2017;71(5):786-794)
-> PHOX2B is a sensitive and specific marker for neuroblastoma.
*p15, line 327: liposarcomatous differerentiation is also described as a rare phenomenon in MPNST (Van Haverbeke C, et al. Pathology 2018;50(4)475-478)
-> Lipoblastic differentiation is also described as a rare phenomenon in MPNST.
*STAT6 expression in dedifferentiated liposarcoma: add references
Creytens D, et al. Appl Immunohistochem Mol Morphol 2015;23(6):462-463
Creytens D. Pathology 2017;49(6):679
-> Above two references added.
*rhabdomyoblastic differentiation is also described as a rare phenomenon in malignant SFT
Creytens D, et al. Int J Surg Pathol 2018;26(5):423-427
-> Rhabdomyoblastic differentiation is also described as a rare phenomenon in malignant SFT.
Reviewer 2 Report
The authors provide a comprehensive well-written review on the diagnostic approach to retroperitoneal sarcomas. Their focus is on pathologic diagnosis. Overall, the report is appreciable and comes with an extensive and updated reference list.
To improve the manuscript - and make it more intriguing to the readership - I would propose the following:
- Since the Title, the authors should specify that their review focuses on pathologic diagnosis (otherwise, the reader could also expect a paragraph on radiologic imaging, for instance).
- Optional: it could be indeed beneficial to include also the other retroperitoneal malignancies (not only retroperitoneal sarcomas). When reading a review like this, my personal (practical) disposition is to expect everything about of retroperitoneal neoplasms (all histotypes).
- The Introduction should highlight the novelties introduced by the new WHO classification (otherwise the paragraphs sounds a little bit like a standard book chapter on the disease)
- Throughout the text, the new developments/tools in the diagnostic approach should be highlighted over what is already known (the most recent references should be discussed further, mentioning the importance of their contribution)
- Equally, all Figures would benefit from an upgrade. The insertion of panels and some information concerning molecular imaging/genetic testing would be really beneficial (otherwise histology and immunohistochemistry alone appear to be a pretty conventional description)
I am confident the manuscript can be improved satisfactorily and then be recommended for publication.
Author Response
The authors provide a comprehensive well-written review on the diagnostic approach to retroperitoneal sarcomas. Their focus is on pathologic diagnosis. Overall, the report is appreciable and comes with an extensive and updated reference list.
To improve the manuscript - and make it more intriguing to the readership - I would propose the following:
- Since the Title, the authors should specify that their review focuses on pathologic diagnosis (otherwise, the reader could also expect a paragraph on radiologic imaging, for instance).
-> The article title was changed.
‘Retroperitoneal Sarcomas: An Update on the Diagnostic Pathology Approach’
- Optional: it could be indeed beneficial to include also the other retroperitoneal malignancies (not only retroperitoneal sarcomas). When reading a review like this, my personal (practical) disposition is to expect everything about of retroperitoneal neoplasms (all histotypes).
-> We appreciate your suggestions. This article does not cover all retroperitoneal malignancies. Other retroperitoneal malignancies such as malignant melanoma, metastatic carcinomas, malignant lymphoma, and germ cell tumors were briefly described in differential diagnoses of selected retroperitoneal sarcomas.
- The Introduction should highlight the novelties introduced by the new WHO classification (otherwise the paragraphs sound a little bit like a standard book chapter on the disease)
-> The last part of introduction was modified as follows.
‘Herein, we review the diagnostic pathology approach to retroperitoneal sarcomas and their updated histological and molecular features introduced by the new 2020 WHO classification of soft tissue tumors.’
- Throughout the text, the new developments/tools in the diagnostic approach should be highlighted over what is already known (the most recent references should be discussed further, mentioning the importance of their contribution)
-> The following sentences were added in molecular testing section.
‘New development tools such as comparative genomic hybridization, gene expression arrays, and next-generation sequencing make important contributions not only to our biological understanding but also to classification, prognostication, and treatment approaches for soft tissue sarcomas.’
- Equally, all Figures would benefit from an upgrade. The insertion of panels and some information concerning molecular imaging/genetic testing would be really beneficial (otherwise histology and immunohistochemistry alone appear to be a pretty conventional description)
-> We replaced Figure. 2b with an image for MDM2 amplification FISH.
Round 2
Reviewer 2 Report
Requests and comments have been addressed quite satisfactorily. The manuscript is worth publishing.
Author Response
- Table 3 WT1 (antibodies against C-terminus) should be included in the differential diagnosis. This is a very useful marker for the diagnosis of DSRCT as its nuclear expression is the surrogate of the EWS/WT1 fusion gene.
-> We included WT1 in Table 3.
2. When speaking about the possibility of the heterologous component in the dedifferentiated liposarcoma (pag. 16), the authors should mention not in general but in the contex of retroperitoneum. Accordingly I strongly suggest to include the following new reference: “Trombatore C, Caltabiano R, Li Destri G., Magro G., Petrillo G., Di Cataldo A (2016) Dedifferentiated Liposarcoma of Retroperitoneum With Extensive Osteosarcomatous Component. Int Surg: May-June 2016, Vol. 101, No. 5-6, pp. 217-221”.
-> We mentioned the possibility of the heterologous component and added
the reference.
“DDLPSs with heterologous osteosarcomatous component should be distinguished from EOS [113].”
- Trombatore, C.; Rosario, C.; Giovanni, L.D.; Gaetano, M.; Giuseppe, P.; Antonio, D.C. Dedifferentiated liposarcoma of retroperitoneum with extensive osteosarcomatous component. Int. Surg. 2016, 101, 217–221, DOI: 10.9738/INTSURG-D-14-00282.1.
4) Pag. 28: when speaking about rhabdomyosarcoma, the authors should include in the list of positive immunomarkers also WT1 (cytoplasmic immunoreactivity obtained with antibodies anti-WT1- N-terminus). In this regard, please cite the new reference: “Magro G, Salvatorelli L, Puzzo L, Musumeci G, Bisceglia M, Parenti R. Oncofetal expression of Wilms' tumor 1 (WT1) protein in human fetal, adult and neoplastic skeletal muscle tissues. Acta Histochem. 2015;117(4-5):492-504. doi:10.1016/j.acthis.2015.02.012”
-> We included WT1 marker and added the reference.
“RMSs show diffuse and strong cytoplasmic expression of WT1 [133].”
- Magro, G.; Salvatorelli, L.; Puzzo, L.; Musumeci, G.; Bisceglia, M.; Parenti, R. Oncofetal expression of Wilms' tumor 1 (WT1) protein in human fetal, adult and neoplastic skeletal muscle tissues. Acta Histochem. 2015, 117, 492-504, doi:10.1016/j.acthis.2015.02.012.